



# Validation of a new spatially-explicit process-based model (HETEROFOR) to simulate structurally and compositionally complex stands in eastern North-America.

Arthur Guignabert[1], Quentin Ponette[1], Frédéric André[1], Christian Messier[2,3], Philippe Nolet[2,3], Mathieu Jonard[1].

[1] Earth and Life Institute, Université catholique de Louvain, Louvain-la-Neuve, Belgium
[2] Centre d'Étude de la Forêt, Université du Québec à Montréal, Montréal, QC, Canada
[3] Institut des Sciences de la Forêt Tempérée, Université du Québec en Outaouais, Ripon, QC, Canada

*Correspondence to*: arthur.guignabert@uclouvain.be



**Abstract**

Process-based forest growth models with spatially explicit representation are a relevant tool to investigate innovative silviculture practices and/or climate change effects because they are based on key ecophysiological processes and account for the effects of local competition for resources on tree growth. Such models are rare, often calibrated for a very limited number of species and rarely in mixed and/or uneven-aged stands, and none are suitable for the temperate forests of Québec. The aim

of this study was to calibrate and evaluate HETEROFOR, a process-based and spatially explicit model based on resource sharing, for 23 functionally diverse tree species in forest stands with contrasting species compositions and environmental conditions in southern Québec. Using data from the forest inventory of Québec, we evaluated the ability of HETEROFOR to predict the short-term growth (5-16 years) of these species at the tree and stand levels, and the long-term dynamics (120 years) of red and sugar maple stands. The comparison between the prediction quality for the calibration and evaluation datasets

showed the robustness of the model performance in predicting individual tree growth. The model reproduced correctly individual basal area increment (BAI) of the validation dataset with a mean Pearson's correlation coefficient of 0.56 and a mean bias of 18%. Our results also highlighted that considering tree position is of importance for predicting individual tree growth most accurately in complex stands with both vertical and horizontal heterogeneous structure. The model also showed a good ability to reproduce BAI at the stand level, both for monospecific (bias of -3.7%, Pearson's r = 0.55) and multi-species

stands (bias of -9.1%, Pearson's r = 0.62). Long-term simulations of red maple and sugar maple showed that HETEROFOR was able to accurately predict the growth (basal area and height) and mortality processes from the seedling stage to the mature stand. Our results suggest that HETEROFOR is a reliable option to simulate forest growth in southern Québec and test new forestry practices under future climate scenarios.



## 1. Introduction

Forest ecosystems are subject to an increased disturbance frequency and intensity caused by global changes leading to large scale mortalities and jeopardizing the ability of forests to sustain the provision of crucial ecosystem services (Trumbore et al., 2015; Seidl et al., 2017; McDowell et al., 2020). It is therefore necessary to account for the high level of uncertainty related to these ongoing and future changes by considering flexible management strategies that increase forest resilience and multifunctionality, in particular those promoting multi-species and uneven-aged stands (Messier et al., 2021; Jactel et al.,

2021; Brockerhoff et al., 2017). However, there is still a lack of knowledge about the ecology of mixed stands as well as guidelines for their long-term management (del Río et al., 2021; Forrester, 2019).

Performing experiments to test the effects of various management strategies and/or future environmental conditions on forests is complicated due to the longevity and slow growth of tree species. Modelling approaches are therefore a useful tool for studying these issues on a long time scale (Pretzsch et al., 2015; Maréchaux et al., 2021; Ruiz-Benito et al., 2020). Of the

many different types of models used in forest management, differing in their structure and complexity depending on the initial objectives (Makela et al., 2000; Porté and Bartelink, 2002), there are three main types: empirical models, process-based models, and hybrid models (i.e. using both empirical and process-based approaches) (Fontes et al., 2010). Empirical models are usually calibrated from descriptive relationships derived from inventory data, and are only suitable for extrapolation to systems and environmental conditions for which they have been parameterized (Fontes et al., 2010). On the contrary, process-

based models (PBM) are more appropriate for investigating innovative silviculture and/or climate change effects as they rely on key ecophysiological processes (e.g., photosynthesis, light interception, respiration, etc.) to simulate forest growth using a set of interdependent sub-models (Bohn et al., 2014; Makela et al., 2000). These PBMs can spatially represent the forest in several ways: at the stand scale, by considering an average tree of the stand; at the cohort scale, by handling the forest as horizontally homogeneous layers; or at the individual tree scale by considering the precise location of each tree in the stand

(Pretzsch et al., 2015). The latter option allows for the effects of local competition for resources on tree growth to be considered and is therefore the most relevant modelling approach for studying forest management strategies in changing environments for uneven-aged and mixed stands (Seidl et al., 2005; Pretzsch, 2022).

The present study is the first step toward the development of a stand-level modelling project that aims to test how contrasting management strategies affect the resilience and multifunctionality of Eastern North-American forests. Long-term simulations

will be carried out by crossing future climate scenarios and disturbances (e.g., windstorms, droughts, biotic outbreaks) with current and alternative management strategies: i) business as usual, i.e., the same management as practiced in the last decades; ii) enriching forests with drought-tolerant species adapted to the expected climate change; and iii) enriching forests with species based on the functional level approach (Aubin et al., 2016; Messier et al., 2021; Aquilué et al., 2021). This latter approach promotes both the functional diversity, (i.e., the diversity of traits represented in the stand) as a means of increasing

adaptation to disturbances through the partitioning of ecological niches, and the functional redundancy, (i.e., when multiple species share similar traits) to ensure the continuity of a function if one species is lost (Messier et al., 2019; Oliver et al., 2015; Mori et al., 2013). To do so, we require a spatially explicit individual- and process-based model in which the main processes, such as light interception, carbon allocation, phenology and water balance, are included. Several forest growth models already exist and have been calibrated for temperate species in eastern North-America, including empirical models, e.g., Artémis

(Power, 2016) and MGM (Bokalo et al., 2013), hybrid models such as TRIPLEX (Peng et al., 2002) and ZELIG-CFS



(Larocque et al., 2011), and process-based models, e.g., SORTIE/BC (Coates et al., 2003) and Forest v5.1 (Schwalm and Ek, 2004). However, neither of these two PBMs, which are the only two PBMs in this region that consider individual tree growth (Pretzsch et al., 2015), correspond to our expectations. Forest v5.1, although very exhaustive regarding the processes integrated, does not consider the spatial representation of each tree. In contrast, SORTIE/BC is spatially explicit but does not integrate water and phenological processes as well as climate change.


Here, we described the parametrisation and validation of the HETEROFOR model, a spatially explicit and process-based model describing individual tree growth based on resource sharing (light and water) specifically developed to simulate complex uneven-aged and mixed stands under various disturbances scenarios (de Wergifosse et al., 2020; Jonard et al., 2020), in structurally and compositionally complex stands in eastern North-America. More specifically, we i) calibrated the model for 23 tree species representing a wide range of functional groups, with species already present in Québec or from southern provenances which could be suitable for planting in the future; ii) evaluated the ability of HETEROFOR to predict the short-term growth (5-16 years) of these species at the tree and stand levels using data from the forest inventory of Québec, and iii) tested if the model could reproduce growth and mortality processes in the long-term (120 years), focusing on red maple and sugar maple, the two main species of Québec's temperate forest.


## 2. Materials and Methods


### 2.1. HETEROFOR

HETEROFOR is a tree-scale and spatially explicit process-based model designed to investigate the response of structurally-complex stands (i.e., uneven-aged and/or mixed stands) to changing environmental conditions and management options (Jonard et al., 2020; de Wergifosse et al., 2020). It is implemented and freely available on CAPSIS (Dufour-Kowalski et al., 2012), a collaborative simulation platform for forest growth and dynamics modelling. HETEROFOR includes various modules and options. An overview of the functioning of the model as well as the description of the carbon-related processes (photosynthesis, respiration, carbon allocation and tree dimensional growth) can be found in Jonard et al. (2020), while the phenology and water balance modules are described by de Wergifosse et al. (2020), the light interception module by André et al. (2021) and the regeneration module by Ryelandt (2019).


In short, HETEROFOR starts by running the phenology routine from meteorological data. It determines for each species the budburst, yellowing and falling dates as well as the daily foliage stage (foliage development stage and green leaf proportion). Furthermore, for the deciduous species, phenology is calculated at the tree scale to account for the extended vegetation period of understory trees (de Wergifosse et al., 2020). The solar radiation intercepted by the trunk and the crown of each tree is then calculated using a ray-tracing approach with the SamsaraLight library of CAPSIS (Courbaud et al., 2003; André et al., 2021).


The gross primary production (GPP) is calculated hourly from the photosynthetically active radiation absorbed per unit of leaf area and the soil water potential using the photosynthesis model CASTANEA also available on CAPSIS (Dufrêne et al., 2005; Farquhar et al., 1980). The net primary production (NPP) is estimated as a fraction of the GPP, depending on the tree dimensions, neighbour competition and air temperature. The NPP is first allocated to foliage and fine roots and, for trees over a given size, to fruits. Remaining NPP is then allocated to structural components (trunk, branches and structural roots) using allometric equations, which derives tree dimensional growth (primarily for growth in height and the remaining for growth in diameter) while considering competition with neighbouring trees (Jonard et al., 2020). HETEROFOR also includes a regeneration module, based on the regeneration library on CAPSIS. Considering the large number of seedlings that may be







present in the understory, explicitly locating all of them would be too time consuming and unrealistic. Instead, the stand is divided into square cells of a given size (10 m x 10 m by default), and seedlings are managed as cohorts of species structured

vertically in several size classes with all individuals within a size class having the same dendrometric characteristics. From the tallest size class to the smallest, the radiation absorbed by each one and transmitted to the next one is computed following the Beer-Lambert's law. Individual growth increment is calculated from the transmittance, and other morphological attributes (crown radius, woody biomass, leaf biomass) are derived from the height or the diameter using allometric relationships. GPP and NPP are calculated for the whole size class using CASTANEA, and compared to the individual biomass increment (i.e.,

individual NPP) to deduce the number of seedlings able to survive with the available radiation (Ryelandt, 2019). Saplings are recruited and spatialized once they reach the recruitment height (12 m by default).

### 2.2. Species

The calibration and evaluation of HETEROFOR were completed for 23 North American tree species, 14 broadleaved and 9 coniferous species (Table 1), including all the major species of managed forests in the Québec temperate forests (*Abies*

*balsamea, Acer rubrum*, *Acer saccharum*, *Betula alleghaniensis, Picea glauca, Pinus strobus*, *Populus tremuloides, Tsuga canadensis*). We also selected species that are present to a limited extent in Québec but which could be suitable for planting in the coming years, mainly northern US species currently at their northern range limit in Québec (e.g., *Acer saccharinum, Prunus serotina, Quercus rubra, Tilia americana,* Fig. S1). These 23 species belong to seven functional groups (Table 1; Fig. S2) according to the clustering of Mina et al. (2022) considering 77 North American tree species and based on nine functional

traits identified as essential for ecosystem functioning and resilience to disturbances (Aquilué et al., 2021; Kühn et al., 2021). Thus, this set of species will allow us to study various types of species mixtures and management scenarios, and particularly those based on functional diversity and redundancy. Mean tree diameter at breast height (DBH) in the selected sites from the Québec forest inventory (see section 2.3.) varied from 11.7 cm for *Betula populifolia* to 21.3 cm for *Tilia americana (*15.9 cm on average for all species)*,* and the range of diameter for a single species varied from 11.6 cm for *Betula populifolia* to

57.4 cm for *Pinus strobus* (34.6 cm on average for all species) (Table 1).

### 2.3. Sites

We selected 200 plots from the permanent sample plots (PSP) of the forest inventory of Québec (MFFP, 2021) to calibrate and evaluate the model. Plot size was 400m² and the time span between two inventories for a given plot ranged between 5 and 16 years (Table S1). In each survey, the diameter at breast height was measured on every tree larger than 9.1 cm DBH

(some smaller ones are still present in the dataset and were considered as recruited trees and kept for calibration and evaluation, see Table 1), whereas tree height was only measured on a subsample of trees (about 15% of the trees). Social status (dominant, co-dominant, intermediate, oppressed) and sun exposure class (from 1 where a tree grows in full light to 4 in the absence of light) of each recorded tree were also indicated in only a few plots.

All plots were selected within the temperate deciduous forest area (latitude <47°; Fig. 1) and based on their species

composition to ensure a sufficient number of individuals of each species of interest. They were also selected to be evenly distributed among the three physiographic regions characterizing this part of Québec (Appalachians, Canadian Shield and Saint Lawrence Lowlands, Fig. 1), which can be distinguished by soil parent material, topography, distribution of permafrost and tree line location (Acton et al., 2015). The plots covered a wide variety of environmental conditions and stand characteristics: mean annual temperature ranged from 0.6°C to 7.1°C, mean annual precipitation comprised of rain and snow





from 919 mm to 1446 mm (average over the 1970-2019 period), mean DBH from 10.3 cm to 27.7 cm, tree density from 325 to 2725 trees ha$^{-1}$ and basal area from 3.5 to 60.6 m² ha$^{-1}$ (Table S1). There were also large variations in soil properties. Soil coarse fraction varied between 0% and 80%, soil depth between 0.12 m and 1 m, and ten different soil textures derived from the USDA textural triangle (Schoeneberger et al., 2012) were represented among all sites with four types dominating (sandy loam, loamy sand, sand, and loam accounted for 85% of the sites; Table S1).


**Table 1. Selected characteristics of the 23 tree species sampled from 200 permanent plots of the Québec forest inventory**

| Species | n trees | n plots | DBH mean (cm) | DBH min (cm) | DBH max (cm) | Vegetation Period [1] (days) | Functional Group [2] |
|---|---|---|---|---|---|---|---|
| *Abies balsamea* | 725 | 78 | 12.6 | 4.3 | 29.0 | - | 1 |
| *Acer rubrum* | 795 | 95 | 14.3 | 4.3 | 52.6 | 157 | 4 |
| *Acer saccharinum* | 161 | 10 | 19.6 | 8.4 | 50.8 | 178 | 4 |
| *Acer saccharum* | 656 | 65 | 16.7 | 3.8 | 47.6 | 157 | 4 |
| *Betula alleghaniensis* | 296 | 45 | 14.0 | 5.2 | 31.9 | 148 | 3 |
| *Betula papyrifera* | 343 | 60 | 14.8 | 8.6 | 54.4 | 156 | 3 |
| *Betula populifolia* | 222 | 22 | 11.7 | 9.1 | 20.7 | 163 | 3 |
| *Fagus grandifolia* | 242 | 24 | 15.4 | 3.1 | 43.2 | 145 | 4 |
| *Fraxinus americana* | 153 | 23 | 15.5 | 4.2 | 40.3 | 173 | 6 |
| *Larix laricina* | 209 | 15 | 14.0 | 7.0 | 28.4 | 146 | 5 |
| *Picea glauca* | 250 | 28 | 15.5 | 6.4 | 44.5 | - | 1 |
| *Picea mariana* | 295 | 19 | 14.0 | 4.7 | 32.0 | - | 1 |
| *Picea rubens* | 310 | 25 | 16.4 | 6.2 | 37.7 | - | 1 |
| *Pinus resinosa* | 257 | 8 | 17.4 | 8.2 | 45.5 | - | 2 |
| *Pinus strobus* | 238 | 25 | 20.9 | 3.7 | 61.1 | - | 1 |
| *Populus grandidentata* | 229 | 18 | 15.9 | 9.1 | 37.4 | 146 | 5 |
| *Populus tremuloides* | 397 | 48 | 15.9 | 8.6 | 46.7 | 145 | 5 |
| *Prunus serotina* | 146 | 19 | 13.1 | 9.1 | 33.7 | 135 | 3 |
| *Quercus rubra* | 246 | 20 | 18.2 | 4.5 | 38.4 | 153 | 7 |
| *Thuja occidentalis* | 445 | 34 | 16.6 | 4.5 | 41.5 | - | 1 |
| *Tilia americana* | 143 | 20 | 21.3 | 8.2 | 52.0 | 153 | 6 |
| *Tsuga canadensis* | 377 | 26 | 18.7 | 4.0 | 41.4 | - | 1 |
| *Ulmus americana* | 130 | 20 | 13.8 | 7.1 | 27.3 | 156 | 4 |

[1] from budburst date to falling starting date
[2] See Figure S2 for more details about functional group characteristics

The 200 stands were classified into five forest types based on their species composition: monospecific broadleaved and monospecific coniferous when a single species accounted for more than 75% of the total basal area of the stand; multi-species broadleaved and multi-species coniferous when, respectively, broadleaved or coniferous trees represented at least 75% of the total basal area of the stand with two species representing at least 25%; and mixed stands, when both coniferous and broadleaved species accounted for more than 25% of the total basal area. In total, there were 32 monospecific broadleaved

stands, 26 monospecific coniferous, 71 multi-species broadleaved, 26 multi-species coniferous, and 45 mixed stands. In addition, species richness ranged from 2 to 12 species per stand and functional richness from 1 to 6 illustrating a high diversity of the selected stands. This provided an adequate dataset to evaluate the ability of the model to simulate growth in structurally-complex stands.

To perform an evaluation with a dataset independent from the one used to calibrate the model (see Section 2.4.), the 200 sites

were split into two datasets of 100 sites (Fig. 1). Therefore, sites numbered 1–100 (Table S1; total n trees = 3754) were used for calibration and sites numbered 101–200 (Table S1; total n trees = 3511) were dedicated to model evaluation, with both datasets being similar in terms of environmental conditions and stand characteristics. This splitting of the sites was also made based on species composition to have at least 100 individuals for each species in each dataset. However, five species were





sparsely represented in the forest inventory plots (*Acer saccharinum, Fraxinus americana, Prunus serotina, Tilia americana,*

*Ulmus americana*). For those species, we chose to use a greater number of individuals for the calibration (for a minimum of around 100 trees for each species) resulting in around 40 trees per species remaining to perform the evaluation. Therefore, these five species were calibrated independently but grouped together as 'other broadleaved' for the evaluation.

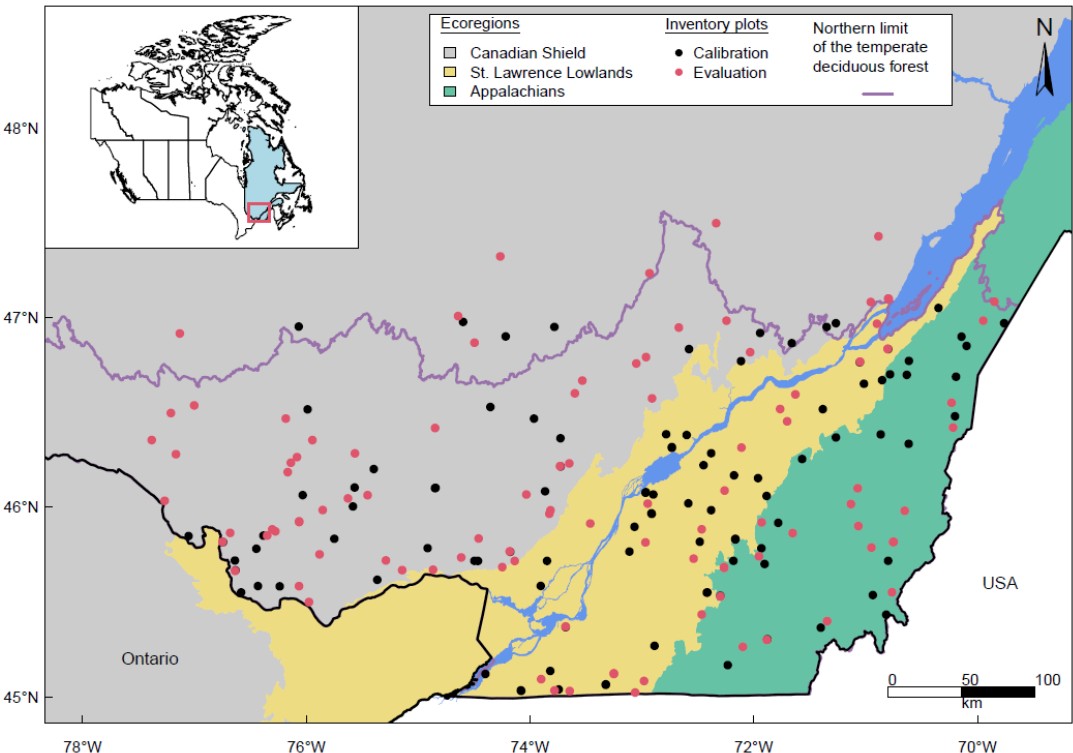

**Figure 1. Location of the 200 selected permanent sample plots of the forest inventory of Québec. The red square in the insert locates**
**the study area within Québec (in blue) and Canada.**

### 2.4. Model calibration

Most of the parameters needed by HETEROFOR are species-specific and are described in Table 2. Values were either retrieved from the literature or fitted with available data when dealing with empirical relationships (see the 'source' column in Table 2). Phenological parameters were calibrated using the Phenological Modelling Platform (Chuine et al., 2013). All

values for each species are given in Table S2. Some other parameters are generic for all species or by species type (broadleaved vs coniferous, deciduous vs evergreen) and are presented in Table S3. The regeneration module has been fully calibrated for only six species so far. Parameters and values for these species are presented in Table S4.

The carbon use efficiency (CUE, kgC kgC$^{-1}$ - corresponding to the NPP to GPP ratio) is a crucial parameter and the only one for which running the model is necessary for the calibration. The CUE is determined for each tree using an empirical

relationship based on tree diameter, a light competition index and temperature, and was computed as:

$$CUE = \alpha + \beta \, dbh + \gamma \, dbh^2 + \delta \ln(LCI) + \varepsilon \, T_{air} + error \qquad (1)$$

where dbh (cm) is the diameter at breast height, LCI the light competition index, $T_{air}$ (°C) the mean annual temperature, and α, β, γ, δ and ε are species-specific parameters. The LCI corresponds to the ratio between the absorbed radiation with and





without neighbouring trees, and ranges from 0 (no light reaching the tree) to 1 (no light competition) (Jonard et al., 2020).
This equation was fitted with data from our first dataset of inventory plots dedicated to calibration (sites numbered from 1 to 100 in Table S1). The NPP was obtained from the two inventories for each tree using the reconstruction mode in HETEROFOR (see Jonard et al. (2020) for detailed information) and then divided by the predicted GPP.

### 2.5. Model evaluation

### 2.5.1. Short-term evaluation: individual tree growth increments

**2.5.1.1. Model initialization**

HETEROFOR requires three different files to be initialized: stand characteristics, soil properties, and meteorological data. The stand characteristics file contains the position of each tree (x, y, z) and its main dendrological characteristics: girth at breast height (cm), total height (m), crown base height (m), height of the maximum crown extension (m), and crown radii in the four cardinal directions (m). The initial observations on each monitoring plot were used for stand input data. However,
only the diameter was available for every tree, and total height for a small subset of the trees. Thus, crown dimensions and total height (when not measured) were estimated using previously calibrated species-specific allometric equations (see Jonard et al. (2020) for the equations and Table S2 for parameters). Tree positions were randomly generated considering the social status of the tree and/or the sun exposure class when available, as well as the size of the trees, ensuring that two trees with a large crown were not positioned too close to each other. Finally, the stand file also includes the longitude, latitude, slope and
aspect of the site.
The model also needs a description of the soil horizons. For each horizon, this file includes the upper and lower limits (m), the coarse fraction ($m^3 m^{-3}$), the bulk density ($kg m^{-3}$), sand, silt and clay contents ($g g^{-1}$), organic carbon content ($mg g^{-1}$), soil pH ($H_2O$) and fine root proportion (%). All of these data were collected from various sources. The organic horizon thickness, sand, silt and clay contents, coarse fraction and soil pH of the horizons were recorded in the forest permanent inventory
database. The description of the soil profile was found in another important ecological inventory of Québec conducted by the Ministry of Natural Resources, *Point d'Observation Ecologique* (POE) (Saucier, 1994). For each inventory plot we selected the closest POE having the same soil type. Bulk density and organic carbon content for each soil type were retrieved from the National Soil Database of the Canadian Soil Information Service (The National Soil DataBase. Canadian Soil Information Service, Government of Canada).
Lastly, meteorological inputs were obtained from the ERA5 global reanalysis (Bell et al., 2021; Hersbach et al., 2020), and provided hourly data of air temperature (°C), soil surface temperature (°C), solar radiation ($W m^{-2}$), rainfall (mm), relative humidity (%), wind speed ($m s^{-1}$) and wind direction (°).

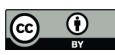

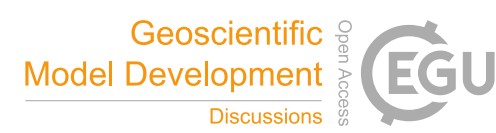
**Table 2: Description of the species-specific parameters used in HETEROFOR (see Table S3 for the generic parameters).**

| Symbol | Description | Units | Source |
|---|---|---|---|
| **Light interception** | | | |
| $k$ | extinction coefficient | $m^{-1}$ | Aubin et al. (2000), Bolstad & Gower (1990), Bréda (2003), Raulier et al. (1999) |
| $SLA_{min}$ | minimum specific leaf area | $m^2\ kg^{-1}$ | Kattge et al. (2020) |
| $SLA_{max}$ | maximum specific leaf area | $m^2\ kg^{-1}$ | Kattge et al. (2020) |
| **Tree dimensions** | | | |
| $hcb\%$ | crown base height | $m^{-1}$ | Calculated with data from USDA - Forest Service (1999) and Power et al. (2012) |
| $Dd$ | crown to stem diameter function ($\alpha$, b, $\gamma$, $\delta$ in Eq. 10 in Jonard et al. (2020)) | $m\ m^{-1}$ | Fitted with data from USDA - Forest Service (1999) and Power et al. (2012) |
| $sh$ | coefficient to shift the mean crown to stem diameter ratio to its maximum | dimensionless | Estimated with data from USDA - Forest Service (1999) and Power et al. (2012) |
| $\Delta dbh$ | default dbh increment | $cm\ yr^{-1}$ | Calculated from the Québec forest inventory data (MFFP, 2021) |
| $\Delta hcb_{max}$ | maximum annual change in the crown base height | $m\ yr^{-1}$ | Calculated from the Québec forest inventory data (MFFP, 2021) |
| $\Delta h$ | height growth function ($\alpha$, b, $\gamma$, $\delta$, $\varepsilon$, $\zeta$, $\eta$ in Eq. 2 in de Wergifosse et al. (2022)) | $m\ yr^{-1}$ | Fitted from the Québec forest inventory data (MFFP, 2021) |
| $V_{tot}$ | tree total volume function (a, b, c in Eq. 5 in Deleuze et al. (2014b)) | $m^3$ | Fitted from biomass data and wood density |
| $V_{stem}$ | tree stem fraction function (d, e, f, g in Eq. 5 in Deleuze et al. (2014a)) | $m^3\ m^{-3}$ | Fitted from biomass data and wood density |
| **Carbon allocation** | | | |
| $b_{leaf}$ | leaf biomass function ($\alpha$, b, $\gamma$ in Eq. 15 in Jonard et al. (2020)) | g OM | Fitted with data from Falster et al. (2015), Ung et al. (2017) and Schepaschenko et al. (2017) |
| $b_{structural\_above}$ | aboveground structural biomass ($\alpha$, b, $\gamma$ in Eq. 26 in Jonard et al. (2020)) | kg OM | Fitted with data from Falster et al. (2015), Ung et al. (2017) and Schepaschenko et al. (2017) |
| $\rho_{stem}$ | stem volumetric mass | $kgC\ m^{-3}$ | Zanne et al. (2009) |
| $\delta_{leaf}$ | leaf relative loss rate | $kgC\ kgC^{-1}\ yr^{-1}$ | Ameztegui et al. (2017), Wright et al. (2004) |
| $\delta_{fr}$ | fine root relative loss rate | $kgC\ kgC^{-1}\ yr^{-1}$ | Coleman et al. (2000), Krasowski et al. (2018), McCormack et al. (2012, 2013) |
| **Respiration** | | | |
| $a_{sapwood}$ | sapwood area function (a, b, c in Eq. 12 in Jonard et al. (2020)) | $cm^2$ | Fitted or retrieved from literature [1] |
| $CUE$ | carbon use efficiency ($\alpha$, b, $\gamma$, $\delta$, $\varepsilon$ in Eq. 1 in this paper) | $kgC\ kgC^{-1}$ | Calibrated using the Québec forest inventory data (MFFP, 2021) |
| **Water balance** | | | |
| $bark\%$ | bark proportion | percentage | Miles and Smith (2009) |
| $\rho_{bark}$ | bark volumetric mass | $kg\ m^{-3}$ | Miles and Smith (2009) |
| $C_{bark\_ll}$ | bark storage capacity in the leafless period (c, d, $R_{min}$ in Eq. 16 in de Wergifosse et al. (2020)) | $1\ mm^{-1}$ | André et al. (2008) |
| $C_{bark\_ld}$ | bark storage capacity in the leaved period (c, d, $R_{min}$ in Eq. 16 in de Wergifosse et al. (2020)) | $1\ mm^{-1}$ | André et al. (2008) |
| $p1_{sw}, p2_{sw}$ | Stomatal response to soil water potential (Eq. 55 in de Wergifosse et al. (2020)) | adimensional | Determined from drought tolerance index of Niinemets and Valladares (2006) |
| **Phenology** | | | |
| $t_0$ | chilling starting date | day of year | Morin et al. (2009) |
| $T_{min}, T_{max}, T_{opt}$ | minimal, maximal and optimal chilling temperatures (optimum chilling model, Eq. 1 in de Wergifosse et al. (2020)) | °C | Calibrated with data from Crimmins and Crimmins (2017) |
| $C_a, C_b, C_c$ | chilling parameters (sigmoid chilling model, Eq. 2 in de Wergifosse et al. (2020)) | adimensional | Calibrated with data from Crimmins and Crimmins (2017) |
| $C^*$ | chilling threshold | °C | Calibrated with data from Crimmins and Crimmins (2017) |
| $F_b, F_c$ | forcing parameters (Eq. 3 in de Wergifosse et al. (2020)) | adimensional | Calibrated with data from Crimmins and Crimmins (2017) |
| $T_{b\_for}$ | base temperature for forcing | °C | Chuine (2000) |
| $F^*$ | forcing threshold | °C | Calibrated with data from Crimmins and Crimmins (2017) |

[1] Anderson-Teixeira et al. (2015), Bond-Lamberty et al. (2002), Bovard et al. (2005), Falster et al. (2015), Hadiwijaya et al. (2020), Hernandez-Hernandez (2014), Hernandez-Santana et al. (2015), Kenefic and Seymour (1999), McIntire (2018), Penner and Deblonde (1996), Quiñonez-Piñón and Valeo (2017), Thurner et al. (2019), Wullschleger et al. (2001)



### 2.5.1.2. Simulations

Stand structure is known to influence light interception and tree growth, and needs to be integrated when modelling structurally complex stands by considering precise tree position and spatial configuration of crowns (Forrester, 2014; Pretzsch, 2022). To investigate the importance of the spatially explicit representation on the prediction accuracy of tree growth increments, we carried out ten simulations per plot, each simulation having a new spatial arrangement of trees. To do this, we ran the semi-random procedure used to locate the trees ten times per plot, resulting in ten stands that were different in terms of spatial arrangement but had the same species composition, tree density and basal area.

### 2.5.1.3. Model performances

The evaluation of the model outputs was performed at the individual tree level focusing on the basal area increment (BAI, cm² yr⁻¹) and height increment (m yr⁻¹) following a two-step procedure: i) comparison of the mean predicted increment (basal area or height) from the 10 simulations to those observed from the forest inventories; and ii) comparison, for each tree, of the best prediction within the 10 simulations with the observed value. This evaluation was done using the hundred plots dedicated to evaluation (plot IDs from 101 to 200; Table S1) to perform an independent evaluation, using 3511 trees with BAI measurements and 508 trees with height measurements.

The evaluation of BAI was carried out for 18 species individually, and the other five (*Acer saccharinum, Fraxinus americana, Prunus serotina, Tilia americana, Ulmus americana*) were evaluated together as 'other broadleaved' (but calibrated independently, see section 2.3.). Regarding tree height increment, sample size for each species was not sufficient to perform a species-specific evaluation. Therefore, we evaluated height increment by grouping all trees as either broadleaved (n=247) or coniferous trees (n=259).

We assessed the accuracy of the model using several metrics. The relative bias identifies underestimated (negative bias) or overestimated (positive bias) overall model predictions, and is calculated as:

$$Bias\ (\%) = \frac{\overline{Pred} - \overline{Obs}}{\overline{Obs}} \times 100 \tag{2}$$

where $\overline{Pred}$ and $\overline{Obs}$ are the means of the predictions and observations, respectively. A paired *t*-test was performed to test bias significance. The root mean square error (RMSE) quantifies the quadratic mean of the differences between predictions and observations and is computed as follows:

$$RMSE = \sqrt{\frac{\sum_{i=1}^{n}(Pred_i - Obs_i)^2}{n}} \tag{3}$$

where $Obs_i$ are the observed values, $Pred_i$ are the predicted values, and $n$ the number of observations.

The strength of the relationship between observations and predictions was investigated with the Pearson's correlation coefficient (r) and with a Deming regression (*mcr* package, Manuilova et al., 2021), which considered errors for both observations and predictions. All the above-mentioned procedures were carried out with the R software version 4.1.0. (R Core Team, 2021).





### 2.5.2. Long-term evaluation: growth and mortality processes starting from regeneration

To evaluate the ability of HETEROFOR to predict growth and mortality processes in the long-term, we conducted 120-year simulations starting with a cohort of 1-year-old seedlings. We focused this evaluation on stands dominated by the two main broadleaved species of Québec's temperate forest, sugar maple (*Acer saccharum*) and red maple (*Acer rubrum*).

The inventory files used to initialize the simulations contained only a regeneration cohort of 20000 one-year-old seedlings of 20 cm in height per hectare, with no overstory trees. For each stand type (red maple or sugar maple), we compared four different compositions of regeneration: i) 100% maple of either red or sugar maple; ii) 75% maple of either red or sugar maple, 25% species A; iii) 75% maple of either red or sugar maple, 25% species B; and iv) 50% maple of either red or sugar maple, 25% species A, 25% species B. Species A and B associated with red maple were yellow birch (*Betula alleghaniensis)* and black cherry (*Prunus serotina)*, and those associated with sugar maple were American beech (*Fagus grandifolia*) and white ash (*Fraxinus americana)*. Average values for southern Québec were used for soil and meteorological inputs required by the model.

We used data from the PSP to compare predicted total standing basal area (m² ha⁻¹) and mean stand height to observed field data. The selected PSP were located in the temperate forest area (latitude < 47°), had a regular structure, had not been disturbed, and were composed of at least 50% of red or sugar maple in terms of basal area. The PSP considered in this sampling may include some PSP used for the calibration or the short-term evaluation. Tree height was only measured on a subsample of trees in the PSP, where selected trees were chosen to represent 3 size classes of the dominant species (largest diameters, around quadratic mean diameters, small diameters). This implies that the heights measured in our sampling plots are almost exclusively maple trees. To be consistent with these characteristics of the PSP dataset, we therefore considered the mean height of maple trees instead of the dominant height in this long-term evaluation. For sugar maple, we also compared the simulations to a dataset from a study of Nolet et al. (2010) about the productivity of even-aged sugar maple stands established following a clear-cut or fire. We used self-thinning relationships to evaluate the ability of HETEROFOR to reproduce the mortality process. Also known as maximum size-density, this relationship describes at maximum stand density the natural process in which tree density per area decreases over time as the average tree size increases (Reineke, 1933). The self-thinning lines from our simulations were compared to those of Andrews et al. (2018) and Lhotka and Loewenstein (2008), which were obtained from data in eastern North-America, and to the dataset of Nolet et al. (2010).

### 3. Results

### 3.1. Short-term evaluation

### 3.1.1. Tree basal area increment

We found that HETEROFOR was able to predicted the basal area increment of the various species (Fig. 2), but prediction accuracy strongly varied between species. The Pearson's correlation coefficient was highly significant (p<0.001) for almost all species ranging from 0.328 for *Q. rubra* to 0.759 for *P. glauca*, except for *B. populifolia* (r=0.23, p<0.05). The bias was less than 25% for 15 of the 19 species, and the RMSE was 5.25 cm² yr⁻¹ on average. BAI was weakly predicted for the trees with the largest BAI of a few species (*A. saccharum*, *B. papyrifera* and *P. resinosa*), but the slope of the regression of observations vs predictions was close to 1 on average (averaged slope = 1.12) and the 1:1 line was within the confidence interval of the regression for 10 of the 19 species (Fig. 2).







**Figure 2. Observed versus mean predicted basal area increment for each species (or group of species) using the evaluation dataset. Each dot represents the mean of the 10 predictions for a single tree, with the error bars indicating the standard deviation. The blue line represents the Deming regression between observed and predicted values, the light blue area is the confidence interval at 95%, and the red dashed line corresponds to the 1:1 line. Model performance is indicated using Pearson's r (p-value: \*\*\* <0.001; \* <0.05), the relative bias (paired t-test, p-value: \*\*\* <0.001; \*\* <0.01) and the RMSE.**



Compared to the mean predictions of BAI performed with the calibration dataset, we observed that modelling performance with the evaluation dataset was slightly less reliable (Fig. 3). With the evaluation dataset, correlations were lower for all species except *A. saccharum,* and predictions were more biased for all species except *Q. rubra* (Fig. 3). On average, the

correlation coefficient between observed and predicted BAI values decreased from 0.678 with calibration plots to 0.568 with evaluation plots, and the bias (in absolute values) increased from 8.8% to 18%. Only a limited number of species showed a strong difference between the two datasets for some of the performance parameters considered (i.e., *B. populifolia* and *Q. rubra* regarding the Pearson's coefficient, *P. resinosa* regarding the RMSE, and *B. populifolia*, *P. mariana* and *P. rubens* regarding the bias). Regressions between observations and predictions were very similar between both datasets (average slope

of 1.15 and 1.12, average intercept of -0.94 and -0.55), and predictions with the evaluation dataset showed an even better relationship for five species with a slope closer to 1 and an intercept closer to 0 (*A. balsamea, L. laricina, P. mariana, P. rubens, P. strobus*) (Fig. 3).

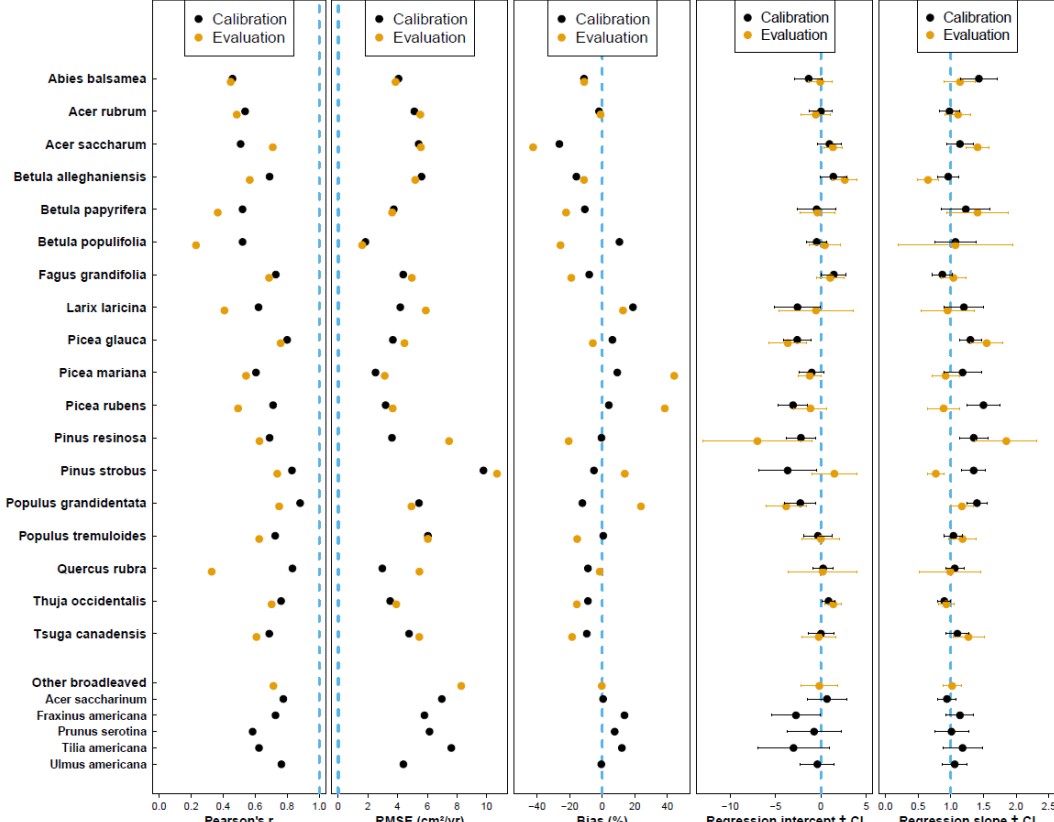

**Figure 3. Statistical parameters (from left to right: Pearson's correlation coefficient, Root mean square error, Relative bias,**
**intercept and slope of the Deming regression) assessing the performance of the model for each species using the calibration (black dots) or the evaluation (yellow dots) dataset. The blue dotted line indicates the best agreement between observations and predictions for each parameter.**

With the random selection of tree positions (10 replicates), we observed a large variation in predictions for a single tree as illustrated by the error bars in Figure 2. The relative difference between the minimum and maximum predicted basal area for
a single tree ranged from 0 to 138%, with 94% of the trees having a difference less than 10%. This relative difference was related to initial tree size, with the largest differences being associated with smaller trees (Fig. S3). Predictions were greatly





improved for all species when focusing solely on the best prediction for each tree (Fig. 4). Pearson's coefficient was always highly significant and ranged from 0.696 (*Q. rubra*) to 0.958 (*P. glauca*). Differences between observations and predictions were less biased for all species with a maximum bias of 24.1% (*A. saccharum*) and a bias <15% for 16 of the 19 species (Fig.

4). Confidence intervals of the regressions were smaller than those obtained from the mean predictions: the slopes were on average similar although the slopes furthest from 1 were much improved (e.g. *P. glauca, P. resinosa*).









**Figure 4.** Observed versus best predicted basal area increment for each species (or group of species) using the evaluation dataset. Each dot represents the best prediction within the 10 simulations for a single tree. The blue line represents the Deming regression between observed and predicted values, the light blue area is the confidence interval at 95%, and the red dashed line corresponds to the 1:1 line. Model performance is indicated using Pearson's r (p-value: *** <0.001), the relative bias (paired t-test, p-value: *** <0.001; ** <0.01) and the RMSE.




### 3.1.2. Height growth increment

Predictions of height increment for the two groups of species were less precise than the predictions of basal area increment (Fig. S4). Considering the mean predictions, the correlation coefficient was 0.304 (p<0.001) for broadleaved species and 0.123 (p<0.05) for coniferous species, and predictions were significantly underestimated in both cases (biases of -33.3% and -31.7%;

Fig. S4). We observed a large variation in predictions for a single tree with a relative difference between the minimum and maximum predicted height ranging from 0.3% to 5.7%. As for BAI, this difference increased as initial tree size decreased (Fig. S3). Focusing on the correlation coefficient, modelling performances of height increment were strongly improved when only considering the best prediction for each tree: Pearson's r increased from 0.304 to 0.729 for broadleaved trees and from 0.123 to 0.718 for coniferous species (Fig. S4). However, the slopes were similar and the biases, which decreased around -20%,

remained significant.

### 3.1.3. Stand basal area increment

The model showed a good ability to reproduce observed mean BAI at the stand level, both for monospecific and multi-species stands, as shown by the regression tests with a slope very close to 1 and the 1:1 line was within the confidence interval (Fig. 5). Moreover, the correlations were strong between observations and predictions (r=0.547 and 0.624) and predictions were

slightly underestimated in both cases.

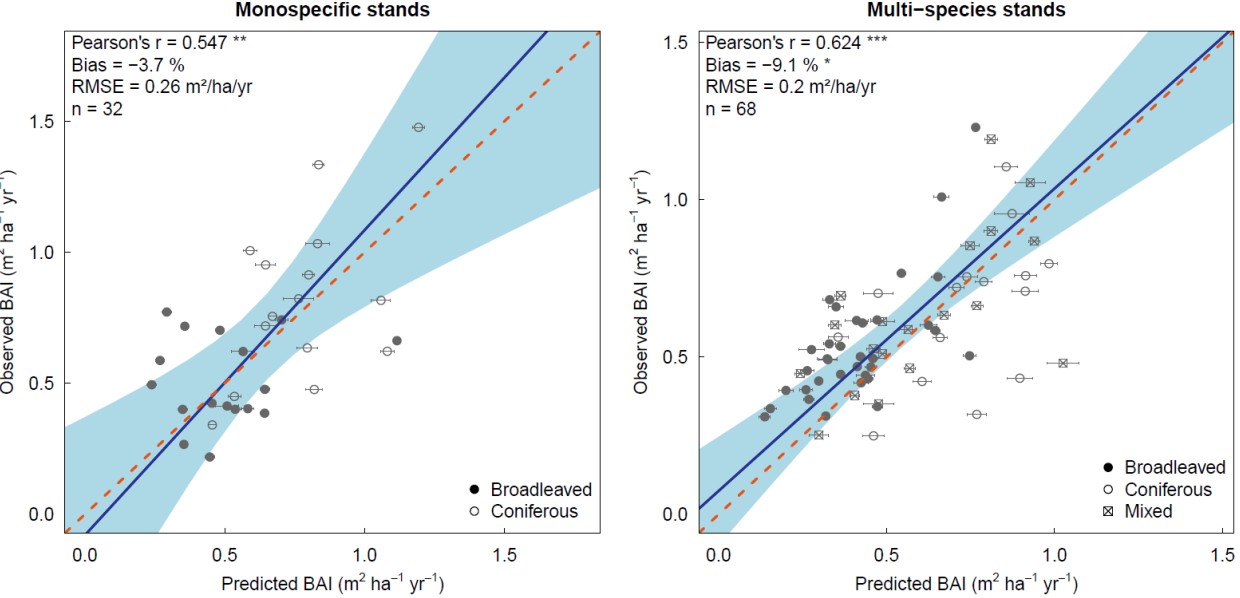

**Figure 5. Observed versus predicted basal area increment at the stand level for the 100 stands of the evaluation dataset, grouped by forest types, monospecific stands (left panel) vs multi-species stands (right panel). The blue line represents the Deming regression between observed and predicted values, the light blue area is the confidence interval at 95%, and the red dashed line corresponds**
**to the 1:1 line. Model performance is indicated using Pearson's r (p-value: *** <0.001, p-value: ** <0.01), the relative bias (paired t-test, p-value: * <0.05) and the RMSE**

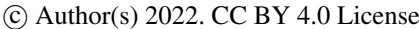
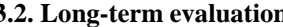

### 3.2. Long-term evaluation

#### 3.2.1. Basal area

The simulated total standing basal area over 120 years is shown in Figure 6 A-B, in comparison with forest inventories. For
the six species used in the simulations, recruitment height was reached between 26 and 36 years. Our results showed that values of basal area for all the different regeneration combinations matched the PSP data at 30 years, suggesting that the regeneration module simulated seedling growth efficiently. Growth from regeneration module outputs to 120 years were very similar for the four simulations of red maple dominated stands. Simulations reached a value of basal area around 35 m² ha⁻¹ at 120 years and agreed with the PSP data over the whole period (Fig. 6A). For sugar maple stands, simulations starting with
100% maple or 75% maple + 25% American beech showed similar basal area and were closely related to the PSP data, but were at the lower range of the values from Nolet et al. (2010). The other two simulations showing a higher growth after 50 years had stands containing 25% of white ash. Both of these simulations were more consistent with the basal area recorded by Nolet et al. (2010) between 70 and 90 years, and reached 36-39 m² ha⁻¹ at 120 years, which is still within the upper range of the PSP data.

**3.2.2. Height**

The simulated evolution of mean height demonstrated a good fit with the PSP data for both maples (Fig. 6 C-D). Height was slightly lower than the average PSP value around 30 years for both species and increased in agreement with the main range of the PSP values over time until height reached 22.1m and 24.6m for sugar maple and red maple stands respectively. Regarding the red maple simulations, mean height was similar among the three stand compositions until 60 years. By the end of the 120
years, height growth was highest in the pure stands (25.6m), followed by the stimulations with 75% of maple (24.8m) and 50% of maple (23.8m) (Fig. 6C). The four simulations were quite similar over time for the sugar maple stands, with final mean height between 21.2m and 22.6m (Fig. 6D). Mean height values from Nolet et al. (2010) range from 13.1m to 24.8m (mean 18.6m) but did not show any consistent trend. As a result, the curves simulated for sugar maple stands did not match their values over the whole time period but were consistent with the overall range (Fig. 6D).

**3.2.3. Mortality**

A visual assessment of the predicted self-thinning lines vs the self-thinning lines of Andrews et al. (2018) and Lhotka and Loewenstein (2008) confirmed the adequacy of the model to reproduce mortality over time for both species, regardless of the initial regeneration composition (Fig. 6 E-F). Compared to the theoretical lines, the predicted density-size relationships at 120 years were excellent for both species, while tree density started to decrease a bit earlier with our simulations than reported vy
Andrews et al. (2018), especially for sugar maple (Fig. 6F). Our predictions also match the values of Nolet et al. (2010) very well, as the predicted curve passes through the scatterplot (Fig. 6F).







**Figure 6. Evaluation of stand basal area (panels A & B), mean height of maple species (panel C & D) and stand density (panels E and F) over 120 years for stands dominated by Acer rubrum (panels A, C, & E) or Acer saccharum (panel B, D & F). The red line represents the self-thinning curves of Andrews et al. (2018) and the red dashed line the self-thinning line of Lhotka and Loewenstein (2008). PSP = permanent sample plots of the forest inventory of Québec. Stand age in the PSP dataset are grouped into 20-year classes, e.g., age class 30 = stands between 21 and 40, except for age class 120 (stands > 100 years). The green dots in captions B, D & F are from Nolet et al. (2010). Acer proportion corresponds to the initial proportion of maple in the regeneration used in the four different simulations: i) 100% maple; ii) 75% maple - 25% species A; iii) 75% maple - 25% species B; iv) 50% maple - 25% species A - 25% species B.**



## 4. Discussion

### 4.1. Ability of HETEROFOR to reproduce individual tree growth

Short-term model evaluation (i.e., 5-16 years) was conducted using forest inventory data from monospecific and multi-species stands, focusing on basal area increment for each species and height increment at the broadleaf/conifer level. Our results

regarding basal area increment are quite consistent with previous studies evaluating HETEROFOR in Europe, yet for a more limited number of tree species (European beech; Sessile and pedunculate oaks). Compared to Jonard et al. (2020), we found a lower RMSE for all species except *Pinus strobus,* as well as a lower correlation between observations and predictions (Pearson's r between 0.23 and 0.76 here, versus 0.63 and 0.83 in Jonard et al., 2020). In another study dealing with the same two species in 36 sites in European, de Wergifosse et al. (2022) evaluated individual tree growth based on girth increment and

found a correlation of 0.58 for sessile/pedunculate oaks and 0.75 for European beech, and a bias lower than 14%. Biases are higher in our study, ranging from -42% to 38% depending on the species, with half of the species showing a bias lower than -14% or 14%. Although our predictions are overall less precise with respect to most indicators, some species are still predicted more accurately than oak and as good as beech (e.g., *Pinus strobus, Thuja occidentalis, Picea glauca*). It should also be noted that these two European studies used many more characteristics (i.e., tree positions, crown dimensions, soil profile, etc.) to

calibrate the CUE and evaluate the model, which definitely increased the accuracy of their model predictions. Comparing our results with other process-based models is difficult as there are only a few spatially explicit PBMs accounting for light-, water- and phenology-related processes (Pretzsch et al., 2015), and where the evaluation of the model performance, when available, is not performed on individual tree growth but mostly on stand level predictions. However, our results are in the same range of biases as the process-based model BALANCE (18% to 47%; Grote and Pretzsch, 2002) as well as those of two hybrid

models evaluated at the individual tree level, SILVA (Schmid et al., 2006; -47% to 70%; Pretzsch, 2002) and ForCEEPS (- 7.3% to 89.9%; Morin et al., 2021). Looking at the stand level, predictions of basal area increment were in good agreement with observed values in both monospecific and multi-species stands. BAI was slightly underestimated in multi-species stands compared to the monospecific stands, but the prediction errors were smaller and the correlation between predictions and observations was higher. The values of the different evaluation metrics at the stand level were consistent with other process-

based modelling studies (Schwalm and Ek, 2004; Forrester et al., 2021; Gonzalez-Benecke et al., 2014, 2016).

Height growth was predicted less accurately than BAI and was underestimated for both conifers and broadleaved species (negative bias of 31.7% and 33.3%), particularly for trees with the highest height increment. This lower accuracy in height predictions is common in forest growth models (Schwalm and Ek, 2004; de Wergifosse et al., 2022; Korol et al., 1995; Strimbu et al., 2017), notably due to the higher potential measurement errors than for diameter. These inaccuracies in height

measurements, which can be estimated within 1m (Jurjević et al., 2020), are present during both initial and final inventories and have a greater impact on predictions in areas like Québec where tree height growth is limited. Nevertheless, although tree height predictions were not perfectly accurate, the height growth over the long-term in both sugar and red maple stands was consistent with observed data. Compared to the study of de Wergifosse et al. (2022) using HETEROFOR, our results are in

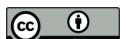



line regarding Pearson's r and the RMSE; however, although they also reported an underestimation of height growth

predictions, the bias was higher in our study (31-33% versus 10-20%).

Considering the large range of stand composition and environmental conditions covered by the plots for most of the 23 species, our evaluation of HETEROFOR demonstrated its ability to accurately predict individual tree growth for these species. The comparison between the prediction strengh for the calibration and evaluation dataset illustrates that our model is robust and can be confidently used to capture the variation in individual tree growth in the temperate forests of Québec. Very few species

showed a clear difference in more than one indicator between the two datasets (*Pinus resinosa*, *Betula populifolia*, *Quercus rubra*). A recalibration of the CUE by combining the two datasets could bring more precision in the tree growth prediction for these species, especially for red pine which is only present in a few sites (3 for the calibration and 5 for the evaluation) with a lack of large trees (>25 cm in DBH) in the calibration sites that could explain the poor prediction of the trees with the largest BAIs.

With more detailed inventory data, two key functions of HETEROFOR involved in carbon allocation processes could also be refined, allowing a better consideration of competition and tree dimension when predicting tree growth. The first key function is the CUE, for which a simplified version based on DBH, LCI and air temperature was used in our study (see Eq. 1 in section 2.4.). However, other predictors based on tree height, crown base height and crown diameter could be added to this equation to account for the effect of tree size and tree shape (slenderness, crown extension) on the CUE (see Eq. 1 in de Wergifosse et

al. (2022)). The second key function is the height growth function, which predicts the annual height growth based on DBH, height, the potential height growth (i.e., potential height increment if all the growth potential is allocated to the primary growth in height and nothing is left for the secondary growth in DBH), the light competition index (LCI) and an error term (standard error of the residuals). As tree positions and most tree height measurements were not available in our dataset, the LCI was not considered when fitting this equation in our study (b=0 in $\Delta h$; see Table S2). However, including the LCI in this function

would allow us to consider that understory trees experiencing high levels of competition for light would generally invest more carbon for height growth than diameter growth (Trouvé et al., 2015).

**4.2. Influence of tree position on model predictions**

Changing tree positions within the stand had a strong effect on tree growth predictions, particularly for smaller trees. This seems logical because the more dominant a tree is, the less it will be affected by the neighbouring trees and therefore by the

change in its position. When focusing on the best prediction for each tree, we showed that the model predictions were greatly improved for all species. This illustrates that considering tree position in process-based tree-level models is necessary to predict individual tree growth most accurately in complex stands with both vertical and horizontal heterogeneous structure (Pretzsch et al., 2015). However, this does not mean that our predictions would match the best predictions if we had the initial tree positions, but we can assume that the predictions would probably be between the mean and the best predictions.

The variation in the position of a tree and the modification of its local environment influence HETEROFOR through three variables: i) the amount of absorbed photosynthetically active radiation, determined from a ray-tracing approach and

consequently impacted by the crowns of neighbouring trees, ii) the specific leaf area, which varies according to the local position of the crown within the canopy, and iii) a defoliation factor, i.e., the leaf biomass of a tree is reduced by defoliation when the available carbon is not sufficient to ensure a normal leaf development (Jonard et al., 2020). However, a random term

in the height growth function (hereafter referred to as height effect) can also have an influence on tree growth between each simulation and may be confounded by the position effect. To disentangle the importance of these two factors (position effect vs height effect) in the simulations, we performed additional simulations on ten sites, considering five different positions per site and five repetitions per position in each site. The five repetitions of each position within each site allowed us to account for only the height effect. We then determined the variation explained by the position and height effects using a linear mixed

model using the girth increment as the response variable and the site and the tree as random factors. The variation explained by the position accounted for 95.05% compared to only 4.95% for the height effect, which confirmed that the position was the most important factor explaining the variations in the predictions among the simulations.

### 4.3. Simulation of maple stand dynamics

The results of the long-term simulations for even-aged stands dominated by red maple or sugar maple showed that

HETEROFOR was able to accurately predict the growth and mortality processes from the seedling stage to the mature stand. Indeed, the evolution of basal area and height growth over time corresponds to the data from the Québec forest inventory, and the self-thinning curves correspond to those previously reported in the same area (Lhotka and Loewenstein, 2008; Andrews et al., 2018).

The calibration and performance of the regeneration module was satisfactory for both maples stand types. The basal area at

the time of recruitment (around 30 years old), i.e., when the saplings are individualized in the model, was very close to that observed in the PSP. However, the seedling height growth was slightly underestimated for the two species and the mortality initiated somewhat early for the sugar maple depending on the self-thinning curve considered. Our results are thus promising regarding the suitability of the model to predict regeneration dynamics and thereby to simulate species composition evolution during the regeneration phase, which is a crucial step of forest dynamics and presents the greatest potential for adapting to

future environmental conditions and unknown disturbance events (König et al., 2022; Kitajima and Fenner, 2000). Further evaluation of seedling growth using long-term regeneration data for maples and other species as well as considering a variety of light conditions in the understory will be of great importance.

Regarding the dynamic of basal area once the trees are recruited, our predictions for pure maple stands of the two species were very close to the mid-range of the PSP. This may be somewhat surprising for sugar maple given the results presented in the

short-term evaluation section, where the model tended to underestimate basal area increment at the individual tree level. However, by looking further into the evaluation site by site, it is evident that the predictions for some sites are very good while others are not, with biases between 1.5% and 62% and Pearson's r between 0.04 and 0.87. The purpose of these long-term simulations was to observe whether the model was able to plausibly reproduce the whole stand dynamics. We only did one simulation with an average soil and climate, which seems to correspond to the mean environmental conditions for the

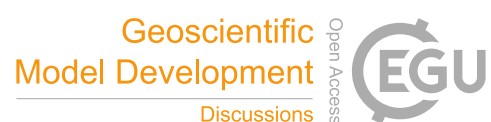

observations. In addition, our results regarding sugar maple growth are quite different from those of Nolet et al. (2010) who observed higher basal areas and mean height, particularly for younger stands. Sugar maples can establish on a wide variety of sites more or less favourable to its growth (Nolet and Boureima, 2009), and several studies and yield tables in Québec, Ontario and North-eastern United States have shown basal area increments similar to ours (Carpentier, 1987; Eyre, 1980; Reed et al., 1994) while others are more in agreement with the values observed by Nolet et al. (2010) (Nolet et al., 2008; Nyland et al.,

2004). In the Nolet et al. (2010) dataset, older stands are characterised by poorer and thinner soils than younger ones, which may explain the same productivity of their sites regardless of age. Therefore, HETEROFOR appears more adequate for simulating sugar maple growth on sites with thin and/or poor soils as well as for mixed stands, as shown by the short-term evaluation, but less suitable when growing in full light on nutrient-rich soils. Finally, the results of the simulations with two or three species were consistent with the diameter growth increment of the different associated species. The white ash is the

species with the highest diameter growth increment (0.45 cm $yr^{-1}$) contrary to the other five species that are within a similar range (0.26 - 0.34 cm $yr^{-1}$; MFFP, 2021), clearly showing that the basal area of the stands containing white ash was significantly higher.

## 5. Conclusion

The purpose of this study was to calibrate and evaluate the performance of the spatially explicit process-based model

HETEROFOR for southern Québec using the plots of the Québec forest inventory representing a large range of environmental conditions and stand structures. Despite the lack of some information needed to initialize the model (tree position, tree height and crown dimensions), our evaluation demonstrated the ability of HETEROFOR to predict both the individual tree growth of 23 species of eastern North-American over the short-term and the stand dynamics of the two main broadleaved species in southern Québec over the long-term. The continuation of this study will include a more detailed assessment of the regeneration

module using long-term regeneration surveys, the calibration of the regeneration parameters for all 23 species, as well as a refinement of the CUE for a few species. However, HETEROFOR can now be considered as an appropriate option to simulate forest growth in Québec's temperate forests and test innovative management strategies under future climate scenarios.

## 6. Code and data availability

The source codes of Capsis and HETEROFOR are accessible to all of the members of the Capsis co-development community.

Those who want to join this community are welcome but must contact François de Coligny (coligny@inrae.fr) and sign the Capsis charter (http://capsis.cirad.fr/capsis/charter, last access: 04 November 2022). This charter grants access on all the models to the modellers of the Capsis community. The modellers may distribute the Capsis platform with their own model but not with the models of the others without their agreement. Capsis4 is a free software (LGPL licence) which includes the kernel, the generic pilots, the extensions and the libraries. For HETEROFOR, we also choose an LGPL license and decided to freely



distribute it through an installer containing the Capsis4 kernel and the latest version (or any previous one) of HETEROFOR upon request from Mathieu Jonard (mathieu.jonard@uclouvain.be). The end users can install Capsis from an installer containing only the HETEROFOR model, while the modellers who signed the Capsis charter can access to the complete version of Capsis with all of the models. Depending on your status (end user vs modeller or developer), the instructions to install Capsis are given on the Capsis website (http://capsis.cirad.fr/capsis/documentation, last access: 04 November 2022).

The source code for the modules published in Geoscientific Model Development (Jonard et al., 2020; de Wergifosse et al., 2020) can be downloaded from https://doi.org/10.5281/zenodo.3591348 (Jonard et al., 2019).

The version of HETEROFOR used for this paper, a user guide of the model, as well as data and scripts used for this study are available from https://doi.org/10.5281/zenodo.7225303 (Guignabert et al., 2022).

## 7. Author contributions

A.G. carried out the calibration, performed the simulations and analysed the model outputs, with support from M.J. and F.A.; A.G., M.J. and Q.P. interpreted the results; F.A. and M.J. developed the model code; C.M. and P.N. provided data; Q.P., C.M. and M.J. acquired financial support for the project; A.G. led the writing of the manuscript with contributions from all authors.

## 8. Competing interests

The authors declare that they have no conflict of interest.

## 9. Acknowledgments

This study was supported by the Fonds de Recherche du Québec (FRQ) and the Fonds de la Recherche Scientifique (FNRS) through the project 'Forests in an uncertain context: comparing contrasting strategies of risk management at the local and regional scales'. A.G. is funded by a postdoctoral grant from the FNRS, and M.J. by the 5-year forest research program 'Accord-cadre de recherche et de vulgarisation forestières' (SPW-DNF). We are grateful to Bert Van Schaeybroeck for

meteorological data acquisition and processing, Louis de Wergifosse for sharing R script about random tree location, and Lana Ruddick for revising the English.

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
