# Peer review of "Validation of a new spatially-explicit process-based model (HETEROFOR) to simulate structurally and compositionally complex forest stands in eastern North-America."

_Geoscientific Model Development, 2022_

## Author Comment (AC1)

We thank both reviewers for taking the time to review this manuscript, for their positive feedback and their comments and suggestions. We have updated the manuscript accordingly (changes highlighted in yellow in the manuscript) and added responses to each comment below.

**Referee #1 - Mats Mahnken**

**General comments**

- The study by Guignabert et al. provides a calibration and evaluation of the process-based forest growth model HETEROFOR for eastern North-American forests. The model is calibrated with data from the forest inventory of Québec and model output on tree and stand level growth is evaluated against an independent data set from the same source. Overall agreement between modelled and observed data on growth and forest dynamics is good, hence the authors suggest that the model is a reliable tool for providing accurate forest growth predictions for Québec, which could be used for testing forestry practices under a shifting climate.

- The study fits well into the scope of GMD. The methods are mostly clearly described, but at some points reproducibility is difficult (see comments below). Title and abstract provide clear outlines of the content of the manuscript, while the language throughout the manuscript is fluent, but imprecise in a few places. The paper is well structured and easy to follow with straightforward figures that help understanding the results. Also, the thematic background and existing literature is presented in an appropriate manner.

- One minor drawback of the study is that it is difficult to judge the exact model performance because the reference data did not contain observations of individual tree positions to be initialized in the model. Taking into account spatial information on tree positions is one of the main strengths of the model. The authors provided a workaround to amend this drawback regarding data gaps in the reference data but the question remains how the model would perform with information of single tree positions. Nevertheless, the study still provides a valuable assessment of model performance as it gives a trustworthy range in which the model performance is expected to be located.

- In summary, the presented study provides a valuable evaluation of an important tool for assessing forest responses under a changing climate for heterogeneous forests. The group of process-based and hybrid models have been shown to represent responses under shifting external conditions better than their empirical counterparts. Additionally, spatially explicit models are needed to describe interactions among individuals in structurally and compositionally complex forests, which are foci of research as they will most likely be more widespread in the future as policy and practice seek to increase resilience to climate change and disturbances by increasing forest complexity. HETEROFOR combines these two approaches, and an evaluation of the model's performance is therefore highly relevant.

**Specific comments**

- The title should contain the term "forest" as it may be unclear for colleagues from other disciplines what a "stand" is
  We added 'forest' to the title.

- A very short description of the mortality module/process, i.e. when/why trees die, would be a useful addition to section 2.1, because understanding how mortality is modelled in HETEROFOR is relevant for the long-term evaluation analysis.
  Mortality is considered differently for regeneration and mature trees. We added the following sentence to the manuscript to describe how mortality is considered for mature trees (L. 106-108): "When a tree does not have enough NPP to support its growth (due to light competition, water stress or ageing), the leaf biomass is reduced inducing a defoliation, which ultimately leads to the death of the tree when the defoliation reaches a given threshold (90% by default)". Regarding the regeneration module, the number of surviving seedlings is obtained as the ratio between the NPP of the whole cohort and the NPP of the average individual which is based on available radiation. This is described on L. 116-117.

- From 2.4 it was not clear whether *CUE* is a constant parameter or variable across time for a given individual with increasing *dbh*. Furthermore, it is not clear where the empirical formula for deriving the *CUE* comes from. Is there a reference for it or is this a specification in the HETEROFOR?
  The parameters of the CUE function are constant, but the CUE itself varies over time as a function of dbh, competition index and air temperature. This function is specific to HETEROFOR and taken from de

Wergifosse et al. (2022), and we have simplified it as we did not have the data to take other predictors into account (e.g. slenderness, crown extension, see L. 435-436 in the discussion). We added the reference to de Wergifosse et al. 2022 when we describe the equation on L. 187.

- L. 186: "The NPP was obtained from the two inventories for each tree using the reconstruction mode in HETEROFOR (see Jonard et al. (2020) for detailed information) and then divided by the predicted GPP." The section would be a clearer to the reader if "[…] to estimate *CUE*." was added to the end of this sentence.
  As suggested, we added 'to estimate CUE' at the end of this sentence.

- L. 197: "Tree positions were randomly generated considering the social status of the tree and/or the sun exposure class when available, as well as the size of the trees, ensuring that two trees with a large crown were not positioned too close to each other": As this is a crucial step of preparing the data for a spatially explicit model, it would be good to have a little more information about the process. How exactly were these covariates considered? Was it possible to check the robustness of this location estimation algorithm with observed data on tree positions? It would also be possible to include this additional section in 2.5.1.2. and reference that section here.
  We have added some sentences to describe the procedure in section 2.5.1.1 of the model (L. 206-214). To test the robustness of the procedure, we tested it on two sites where the position of the trees is known: a beech stand in Louvain-la-Neuve, Belgium, and a mixed oak-beech stand in Baileux, Belgium. By comparing the predictions with known positions with those from the random positioning procedure, we can conclude that the procedure used is robust and, at least for these two specific sites, the best prediction for each tree matches the prediction with known positions very well.

- Louvain-la-Neuve – beech stand:

[Figure]

- Baileux – oak-beech stand:

[Figure]

- L. 254: "We focused this evaluation on stands dominated by the two main broadleaved species of Québec's temperate forest, sugar maple (Acer saccharum) and red maple (Acer rubrum)." Why only broadleaved species? Do you assume similar model performances for coniferous and mixed stands?
  These two species are not only the two main broadleaved species, but also the two major species of the Québec's forest considering all the species together. We removed the term 'broadleaved' in the quoted sentence. We did not perform this long-term evaluation for conifers because of the limited data available on seedlings growth. Indeed, the regeneration module is currently only calibrated for a few broadleaved species (sugar maple, red maple, American beech, white ash, yellow birch, black cherry). We expect the quality of predictions to be similar for coniferous and broadleaved species, as well as for mixed stands, as shown in Figure 5 where basal area increment of mixed stand is even better predicted than that of monospecific stands.

- "Predictions were greatly improved for all species when focusing solely on the best prediction for each tree." (l. 306). "However, this does not mean that our predictions would match the best predictions if we had the initial tree positions, but we can assume that the predictions would probably be between the mean and the best predictions." (l. 438): These sentences underline the need for an evaluation with observed stem positions. Is the approach presented for producing the tree positions the standard application procedure in HETEROFOR?
  The approach presented here to determine tree position is not a standard procedure of HETEROFOR, but only used in this specific case. Indeed, HETEROFOR has been so far calibrated, evaluated, and used on sites where tree positions were known, in particular thanks to the long-term monitoring networks of forest ecosystems in France (RENECOFOR) and Belgium (REWASEF). For Québec, it was not possible to find this type of site, with at least 2 inventories on stands where the stems had been mapped. We therefore adopted another approach (semi-random positioning procedure, 10 repetitions) to evaluate the model.

- The language is imprecise in some places. I suggest that the authors go through the manuscript once more to check for words and phrases that are not specific, e.g. "not positioned **too** close to each other" (l. 199); the use of the terms "accurate" and "precise" (e.g. l. 416), which should not be used synonymously"; "which decreased around -20%" (l. 324): decreased **to** or **by** -20 %?
  Thank you for pointing this out. We have modified the text accordingly (decreased to L. 337; accurate instead of precise, L. 330, 400 and 429). "Not positioned too close to each other" was removed as the positioning procedure was explained in more details.

- Potential reasons for observed-predicted offsets potentially resulting from the model structure are not well discussed, e.g. "However, the seedling height growth was slightly underestimated for the two species and the mortality initiated somewhat early for the sugar maple depending on the self-thinning curve considered" (l. 461), could this be traced back to any specific source of uncertainty, be it input data, model parameter or model structure uncertainty?
These offsets can be linked to the characteristics of the data used to calibrate the regeneration module. We have added a sentence to address this point, L. 475-478.

- L. 462: is it really valid to say "results are thus promising regarding the suitability of the model to predict simulate regeneration dynamics and thereby to species composition" if the species composition was fixed at the beginning of the simulation without natural species assembly taken into account and no feedback from mature trees, as done here?
We agree with your comment. For the moment and therefore for this study, we do not take into account the whole regeneration cycle (seed production, seed dispersal, germination, establishment) but only the seedling stage, yet the description of the other stages of the regeneration cycle is under development in the model. We have reformulated the sentence to be more in line with what we have done in this study, as follows: "results are thus promising regarding the suitability of the model to simulate seedling growth and mortality processes and thereby to test the introduction of new species" (L. 479-480)

**Technical corrections**
- L .85: the sentence "HETEROFOR includes various modules and options." could be omitted as it does not provide any relevant information to the reader.
The sentence was removed as suggested
- L. 205: "another important" can be skipped
Corrected
- L. 275: predicted -> predict
Corrected
- L. 292: to avoid confusion with model parameters, the term "performance parameters" could be exchanged with "performance metrics" as the term metric is already being used in the section on the methods and in the discussion
Corrected
- L. 342: "very" could be omitted
Corrected
- L. 364: "vy" -> "by"
Corrected
- L. 380: "quite" could be omitted
Corrected

**Referee #2**

This paper describes the calibration and validation of an individual-based forest model. The authors thoroughly tuned the key parameters (a total of 28) for 23 North American tree species using data from around 100 plots of the forest inventory of Québec, and used another 100 plots to validate model predictions. This is a hard work (at least to me!), tedious, but informative. I have more critics to the model developers than to the model users. The topic falls in the scope of this journal (Geoscientific Model Development) and the paper is generally well written.

I just have some questions to ask the authors and hope they can clarify them in a revised version.

1. From the parameter table (Table 2), I see the growth of tree height, diameter, and crown are set as "parameters". I am wondering that if the model use allometry equations to describe tree size and structure. It seems not. Otherwise, the authors should tune allometry parameters and then use allometry equations to calculate tree growth.

The growth of tree height, diameter and crown are not described directly based on parameters but are determined based on algorithms and functions including several parameters. Growth in height and diameter are derived from the available NPP after allocating the carbon to the functional organs (leaves, fine roots and fruits). All these processes about tree dimensional growth are described in detail in Jonard et al. (2020): Jonard, M., André, F., de Coligny, F., de Wergifosse, L., Beudez, N., Davi, H., Ligot, G., Ponette, Q., and Vincke, C.: HETEROFOR 1.0: a spatially explicit model for exploring the response of structurally complex forests to uncertain future conditions – Part 1: Carbon fluxes and tree dimensional growth, Geosci. Model Dev., 13, 905–935, https://doi.org/10.5194/gmd-13-905-2020, 2020

And, these growth rates should be age/size dependent. Do users need to tune them separately?

The carbon use efficiency (CUE) function as well as the allometric equations used to allocate carbon to tree organs and to describe dimensional growth are size dependent. Once calibrated, these functions do not need to be tuned by users for specific case studies.

Another question, since the NPP calculation and tree size growth are estimated separately, does the model calculate carbon cost of the tree growth (i.e., if an individual tree has enough NPP to support its growth)?

The CUE concept allows converting the GPP into NPP and therefore implicitly accounts for carbon losses due to maintenance and growth respiration. While the growth respiration is proportional to NPP, the maintenance respiration depends on temperature and on the living biomass whose proportion changes with tree size and competition conditions.
And if a tree does not have enough NPP to support its growth (due to light competition, water stress or ageing), the leaf biomass is reduced inducing a defoliation, which ultimately leads to the death of the tree when the defoliation reaches a given threshold (90% by default).
We clarified this in the manuscript on L. 100 and L. 106-108.

2. Apparently, there are many phenomenological equations that are used to link different variables. Are these equations species and location specific?

These equations are species-specific, but not site-specific because the objective was to have a model that was accurate over the entire temperate forest of Quebec. Nevertheless, it is possible for any other user to recalibrate these equations with site-specific data if necessary.

3. For the long-term predictions (more than one hundred years), there are stochastic demographic processes that change the forest structure (e.g., the random dispersion of seeds, mortality, and regeneration and the positions where the events happen). How does the model choose one case in these random events? Does it affect predicted forest structure and dynamics? If yes, do model users need to make an ensemble of model runs and see how a forest develops in a stochastic system?

There is not much stochasticity in the model. Seed dispersal is not currently considered in HETEROFOR, and the composition in the regeneration is predetermined at the beginning of the simulation. Stochastic aspects can be found in: i) the positioning of the trees: once the cohorts have reached the recruitment height, the trees will be randomly placed within each cell; and ii) in the height growth function, which

considers a random term (corresponding to the standard error of the residuals). We observed that these stochastic aspects have limited influence on the model predictions and that the simulations are repeatable.

4. When trees grow, their crowns get larger and thus some of them must be left behind or crown shapes must change (usually we call it "crown packing"). Does this affect the parameters of a tree? Or the parameters for a species apply for all conditions and life stages?

Two different approaches are implemented in HETEROFOR to simulate the crown extension considering either competition with neighbouring trees or not:
- a distance-independent approach, where local competition is not considered and changes in crown dimensions are derived from dbh or height increment based on empirical relationships
- a distance-dependent approach, describing the changes in crown dimensions based on the competition with the neighbouring trees. In this case, the space around a target tree is divided into four sectors according to the four cardinal directions and at most one competing tree is identified for each sector. The changes in the four crown radii (corresponding to the four cardinal directions) are calculated based on crown radii at equilibrium ($cr_{eq}$), which are estimated by considering the competitive strength of the target and competing trees. For a given direction, $cr_{eq}$ is calculated based on the potential (free-growth) crown radius of the target tree and its competitor, the distance between the two trees, and a crown overlap ratio accounting for the capacity of a tree species to penetrate in neighbouring crowns.

More details and associated equations can be found in Jonard et al. (2020).
In this study, we used the distance-dependent approach. This information was added to the text L. 104-106.

5. There have been many efforts to reduce the complexity of this type of forest models, such as the model of Perfect Plasticity Approximation (Strigul et al. 2008). A little discussion of why we still need the tree spatial position-explicit model is necessary.

In the introduction, we described the different types of models, and why tree-level models are appropriate to study structurally-complex stands in changing environments. We have added some details on why considering spatial arrangement was important for that, as well as the two suggested references, L. 50-52. In the discussion, the effect of position is already discussed in section 4.2, which we believe is sufficiently detailed as it is.

6. I think it would be valuable to publish the tuned parameters for these 23 species as appendix or supplementary material.

The tuned parameters for the 23 species are presented in supplementary material (see table S2)

Maybe, I should not ask the authors of a model calibration paper question about the model development. I still think it is necessary to explain that, since the calibration processes are way complicated for me, and it is good to let the readers know the specific value of this model.

As for PPA model, please refer to:

Strigul et al., 2008. Scaling from trees to forests: tractable macroscopic equations for forest dynamics

Purves et al. 2008. Predicting and understanding forest dynamics using a simple tractable model.

---

## Referee Report (RR1)

The revision of the manuscript gained reproducibility and quality. The authors addressed specific comments in an excellent manner and resolve the minor/major issues on language and reproducibility. This study advances our understanding of ecosystem dynamics in heterogeneous forest stands, which is urgently needed. I have no further comments or requests to the authors.

Thank you very much for the work.